# COSMOtherm as an Effective Tool for Selection of Deep Eutectic Solvents Based Ready-To-Use Extracts from Graševina Grape Pomace

**DOI:** 10.3390/molecules26164722

**Published:** 2021-08-04

**Authors:** Manuela Panić, Veronika Gunjević, Kristina Radošević, Marina Cvjetko Bubalo, Karin Kovačević Ganić, Ivana Radojčić Redovniković

**Affiliations:** Laboratory for Cell Culture Technology and Biotransformations, Department of Biochemical Engineering, Faculty of Food Technology and Biotechnology, University of Zagreb, Pierottojeva ulica 6, 10000 Zagreb, Croatia; mpanic@pbf.hr (M.P.); vgunjevic@pbf.hr (V.G.); kristina.radosevic@pbf.unizg.hr (K.R.); mcvjetko@pbf.hr (M.C.B.); kkova@pbf.hr (K.K.G.)

**Keywords:** COSMOtherm, ready-to-use extracts, grape pomace Graševina, deep eutectic solvents

## Abstract

The aim of this work is to develop an industrially suitable process for the sustainable waste disposal in wine production. The proposed process involves the development of an environmentally friendly method for the isolation of biologically active compounds from Graševina grape pomace according to the green extraction principles, in order to obtain a ready-to-use extract. In this process, deep eutectic solvents (DES) were used as extraction solvents. Aiming to save time in selecting the optimal DES that would provide the most efficient Graševina pomace polyphenols extraction, the user-friendly software COSMOtherm was used and 45 DES were screened. Moreover, the prepared extracts were chemically and biologically characterized to confirm their safety for human application. Computational and experimental results proved the applicability of COSMOtherm in the selection of the optimal DES for the environmentally friendly preparation of the ready-to-use extract from Graševina grape pomace with expected application in the cosmetic industry.

## 1. Introduction 

Polyphenols are secondary plant metabolites, currently recognized by the scientific community and the general public as an essential part of human nutrition due to their exceptional antioxidant and antimicrobial properties, as well as other biological activities (antimutagenic, anticarcinogenic, anti-inflammatory). The role of these compounds in the prevention of cardiovascular diseases has been confirmed by numerous scientific in vitro and in vivo studies, as well as epidemiological studies [1,2,3,4]. In addition to the use of unprocessed plant material, agro- and food-industry waste could be exploited for polyphenols extraction as part of a biorefinery approach [5]. It is clear how the further utilization of these wastes as raw material for the isolation of bioactive components can increase the economics of production, firstly by reducing the amount of waste, and secondly by adding value. Grape pomace is one of the food industry by-products that is extremely rich in these biologically active compounds. Moreover, this valorization option could add value to the whole wine making process [3,6,7]. Wine production is a significant agricultural sector in the Republic of Croatia. In the 2018 wine year, about 123.1 t of grapes (the largest share of which is Graševina varieties) or a total of 984,730 hL of wine were produced [8,9]. Waste from wine production alone accounts for 20–25% of the total mass of grapes used in production, which, according to FAO/WHO, is up to 5–10 million t at the global level and the estimate at the Croatian level is over a 15,000 tons of solid waste [10,11,12]. The improper disposal of this waste poses a significant environmental risk, polluting soil and water. In view of the many environmental problems, the awareness of wine waste as a high-value by-product and its potential as a raw material has now increased significantly, and the solution to this problem lies in its qualitative recovery [5]. Most of the wine production waste consists of grape marc, which contains the skins and seeds obtained after pressing and the stalk of the grapes and grape seed. The epidermis, the seeds and the stalk are a rich source of polyphenolic compounds. In addition, grape seed oil contains polyphenolic antioxidants as well as a high content of essential fatty acids, mainly linolenic and oleic acids, and other antioxidants such as tocopherol and *β*-carotene, but also phytosterols, while in the grape skin mainly tartrates (tartaric acid) remain [13]. Environmental preservation and industrial processes and product safety are gaining substantial attention by the scientific community and among the general public. Therefore, industry is facing a new challenge: to establish more environmentally friendly and safer processes while keeping their production economically viable. For the food and pharmaceutical industries involved in the preparation of products containing secondary plant metabolites intended for human consumption, ready-to-use extracts could be the perfect solution. The preparation of this type of extract does not require any recovery steps, and thus brings time and cost reduction. The ideal solvents for their preparation could be deep eutectic solvents (DES). In particular, these solvents have been reported to enhance the biological activity and the stability of the biologically active compounds such as polyphenols, in addition to having a GRAS status [14,15,16,17].

The advantages of DES as extraction media are already known and proven, meaning that they are promising solvents for the extraction of the highest content of polyphenols and are safe to use [17,18,19,20]. DES are a mixture of a hydrogen bond donor (HBD), which can be sugars, polyols, amino acids, amides and a hydrogen bond acceptor (HBA) such as choline chloride in a defined stoichiometric ratio. DES fully complies with the green chemistry principles, and the cost of DES is comparable to that of conventional solvents. DES are chemically and thermally stable, non-volatile, non-flammable, non-toxic and biodegradable. In addition to good economic and environmental properties, DES are tailor-made solvents, and it is possible to design solvents with the specific physicochemical-properties for a particular purpose [21,22]. However, experimentally, it is expensive and time consuming to find the optimal DES for desired compounds extraction. To save time and money and reduce the number of experiments to find the most suitable solvent among 10^6^ possible combinations, COSMOtherm software can be used. Currently, the Conductor-like Screening Model for Real Solvents (COSMO-RS) is considered one of the most accurate ab initio computational methods available for ranking solvents [23]. Although it was not developed for DES, it is possible to create your own database using the TmoleX software. The input for TmoleX is the molecular structure (2D, 3D, or SMILES) and, after optimization, the geometrically and energetically optimal molecule can be created. This combination of these two softwares allows computational search of DES for polyphenol extraction.

In this work, COSMOtherm was used as a tool to select and design optimal DES for Graševina grape pomace polyphenols extraction, aiming to prepare ready-to use extract rich in polyphenols. After designing the optimal DES for extraction, grape pomace extracts were prepared and analyzed. According to the extraction efficiency, betaine:glucose was selected as the optimal DES for polyphenols extraction. Furthermore, antioxidant activity was evaluated by ORAC assay. Finally, grape pomace extracts were assessed by examining their in vitro biological activity on human keranocyte cell line (HaCaT) to test the possibility of applying betaine:glucose grape pomace extract in the cosmetic industry.

## 2. Results and Discussion

### 2.1. DES Selection for Optimal Extraction of Polyphenols from Grape Pomace 

The aim of this work is to prepare ready-to-use extracts rich in biologically active compounds (polyphenols) from wine production waste (Graševina grape pomace) using DES as an extraction solvent. This approach is valuable because the extracts obtained could be directly used as ready-to-use extracts in food, cosmetic and pharmaceutical industries without the need for complex and expensive purification processes, having the DES GRAS status [24]. Moreover, the application of these solvents to the extraction of biologically active compounds could lead to selective extraction of compounds, resulting in a new product that could not be obtained with conventional solvents [17,25]. The search for the ideal DES for a given system has been so far guided by an empirical trial-and-error approach. Thus, to save time and money, and to reduce the experimental effort in searching for the most suitable solvent, we have herein applied in silico method–user-friendly software COSMOtherm [23,26]. Subsequently, the prepared extracts were chemically and biologically characterized. 

The first step was to select the optimal DES for grape pomace Graševina polyphenols extraction. For this purpose, the solubility of catechin, the most abundant polyphenol in Graševina (21.04 mg L^−1^ [4]), was evaluated in 46 different DES using the COSMOtherm software’s option-activity coefficient calculation. A similar computational approach for the design and optimization of DES was used, for example, by Zurob et al. (2020) [27] in the ultrasound-assisted extraction of hydroxytyrosol from olive leaves.

Input to the software is chemically and energetically optimized polyphenol, as well as HBA and HBD for DES preparation in TMoleX software. Afterwards, activity coefficient calculation at infinite dilution of polyphenols in 46 different DES with 30 % (*w*/*w*) of water (when hydrophylic DES were used) at 60 °C. All hydrophylic DES were prepared with 30% of water, as these DES exhibited the best physicochemical properties for extraction [28]. The first software output that can help in DES design is the σ-profile of the polyphenol, catechin, which gives us information about the molecular polarity (Figure 1).

The broad peaks around 0 e/A^2^ × 10^−2^ shows the apolarity of catechin, and the peaks from 0.025–0.01 show that this polyphenol also includes polar regions. From these results, it is expected that the best DES to dissolve target polyphenols are the polar ones. The second valuable COSMOtherm output in DES designing is the activity coefficients calculation (γ∞) at infinite dilution of the polyphenol in DES (Figure 2). The molar ratios of the DES forming compounds in each tested DES are listed in Table 1. The lower activity coefficient of the tested HBA and HBD implies higher compound solubility [23,26,29]. 

Figure 2 shows that there are ten different HBA with 20 HBD that can form different types of DES—from apolar to polar. The highest values of ln γ are found in DESs formed of menthol or thymol as HBA and fatty acids, coumarin and thymol as HBD, and in these DES catechin is not soluble. Ln γ in other DES is negative, which means that in polar DES the tested polyphenol is soluble. The lowest ln γ value is obtained when combining betaine with ethylene glycol (BEG), glucose (BGlc) and sucrose (BScu) in molar ratio 1:1 and the addition of 30% of water. According to the ln γ value, these DES were selected for grape pomace extract preparation. In order to test the software’s reliability, the grape pomace extract was also prepared in Ty:C10 (1:1), since this DES was found to be the most unsuitable for the tested polyphenol extraction. From the obtained results and the literature data, it can be concluded that the physicochemical properties of the tested DES do not have a strong influence on the total polyphenols’ yield. For example, the pH-values of BGlc and BEG are quite similar (6.64 and 6.86, respectively), but the total polyphenols content differs significantly (26.07 and 8.72 mg g_dw_^−1^ of pomace, respectively). The same effect was observed for DES’ polarity influence on the polyphenols’ yield [30,31]. Moreover, according to the website of Merck, Germany (20 June 2021), the estimated price of BGlc is close to the organic solvents currently used in the industry (70.22 € kg^−1^) and is therefore suitable for application. For BEG, the price is 137.5 € kg^−1^ and for BScu 164 € kg^−1^.

The software prediction in selecting an ideal DES for extraction was further evaluated by the chemical characterization of the obtained extracts (Table 2). 

The extract obtained with BGlc contained the highest total polyphenols’ yield (26.07 mg g _dw_^−1^ of pomace), analogous to that obtained with EtOH as the reference solvent (26.06 mg g _dw_^−1^ of pomace). This highlights the possibility of DES to replace conventional solvents, such as ethanol, in the extraction process. Although ethanol is still considered a green solvent, it is not only a taxed solvent, but also requires the use of explosive atmospheres (ATEX) and antiflame equipment [32]. Therefore, replacing ethanol with a non-volatile and non-flammable solvent that has the same extraction efficiency is highly desirable. The other DES selected by COSMOtherm and experimentally tested gave the total polyphenolic yields in the following order: BEG > BScu > TyC10. The total polyphenol yields values ranging from 6.64 to 26.06 mg g^−1^_dw_, indicating a wide variation in DES extraction efficiency. Based on these data, we can conclude that activity coefficient calculation is a good parameter for the DES design, since extraction efficiency with TyC10, which has the most positive ln γ value, is the lowest, whereas in the case of solvents with the most negative ln γ, the efficiency is higher. 

Furthermore, the polyphenolic profile was determined by HPLC-DAD (Table 1). The predominant compounds in all the prepared extracts were found to be flava-3-ols. Among these compounds, catechin was the most abundant. Moreover, five different procyanidins were also identified and quantified. In general, white grape pomace is particularly rich in flavan-3-ols, especially in catechin, hence the obtained results are in accordance with the literature [6,7]. Quite similar catechin content was found in the ethanolic and DES extracts. Nevertheless, BGlc and BScu allowed the extraction of a wide range of different flavan-3-ols, including various procyanidins. Apart from flavan-3-ols, the flavonoid rutin was found in all extracts. DES BEG provided the highest rutin extraction yield. The TyC10 extract was also analyzed by HPLC-DAD and showed the presence of procyanidin B1, which was the only phenolic compound detected in this extract. These results confirm once again the possibility of selective compounds recovery when DES is used.

The chemical characteristics analysis of the prepared extracts led to selection of BGlc as the solvent for the ready-to-use extract preparation from grape pomace Graševina. This extract was further characterized biologically.

### 2.2. Biological Evaluation of Prepared Extracts

As mentioned above, this work aims to prepare ready-to-use polyphenols’ enriched extracts for application in the cosmetic industry. Before industrial application, it is important to evaluate their biological activity and more importantly, their safety. It is known that polyphenols from plants possess many different biological activities, among which the most studied ones are antioxidant and antitumor activity [33]. Based on current literature data, the difference between biological activity determined for extracts prepared using different DES compared to conventional extraction is observed [15,28]. Apart from the green character of DES, which is the reason why this solvent was chosen as the extraction medium, the fact that the extracts prepared in DES could have a different and/or enhanced biological activity than pure polyphenols, is the second attractive advantage for the industrial application of DES based plant extracts. In this research, the antioxidant activity of BGlc extract was determined by ORAC assay, while the cytotoxicity was determined by in vitro assay on the HaCaT cell line. For any product in the cosmetic industry, it is crucial to evaluate its cytotoxicity on human skin to ensure safe use. Therefore, as the model of human skin HaCaT cell line was used. For both antioxidant and antiproliferative activity, the results were compared between the BGlc and conventional extraction. 

The determined ORAC values of the prepared extracts were 173.60 ± 0.82 μmol TE g_dw_^−1^ for GPBGlc and 164.59 ± 13.77 μmol TE g_dw_^−1^ for GPEtOH. It can be concluded that the DES ORAC values correlate with total phenolic content as expected (Table 1), and are in agreement with the results on phenolic grape extracts obtained with DES [28]. The ORAC value of the DES based extract is slightly higher, which may be due to the impact of DES itself, that is, from DES forming compound betaine, which alone possesses antioxidant activity [34]. Previously, ORAC values of DES were determined in papers published by Radošević et al. (2018) and Mitar et al. (2019) [35,36].

The effect of the extracts prepared with BGlc and aqueous EtOH on cell proliferation was evaluated by CellTiter 96^®^ AQ_ueous_ One Solution Cell Proliferation assay. The obtained extracts were applied to the cells at different volume ratios 0.5–5% (*v*/*v*) over 72 h (Figure 3). 

HaCaT cells showed higher cell viability depending on the applied extract volume ratio (Figure 3). It is clearly seen that the growth of HaCaT cells stimulates up to 40% depending on the volume ratio and the type of extract. The higher the volume ratio of extract, the higher the stimulatory effect. Comparing the DES based extract with the ethanol based one, a 15% higher stimulatory effect on the cells was observed when treated with DES based extract (5%, *v*/*v*). It can be concluded that the stimulatory effect depends proportionally on the polyphenols concentration. The observed results with polyphenols have been reported previously [37,38]. In Panic et al. (2020) [17], a positive effect on the growth of HaCaT cells treated with DES-based extracts was also observed, especially with polyphenol-rich extracts of cocoa by-products in BGlc and ChCl. There were also differences in cell growth depending on the type of DES, and BGlc extract stimulated HaCaT growth more than the ChCl-based one. From the presented results, based on the antioxidant capacity and the positive effect on the keratinocytes’ growth, it can be concluded that the prepared ready-to use extract can be used in the cosmetic industry, where its use could be beneficial from two perspectives: the protection of skin cells against oxidative stress as well as the stimulation of cell growth, for example, in skin regeneration. Finally, its antioxidant property could also be beneficial as a stabilizer of cosmetic formulations for longer shelf-life.

## 3. Materials and Methods

### 3.1. Chemicals and Materials

All chemicals and standards were purchased from Sigma (St. Louis, MO, USA) and ChemFaces (Wuhan, China).

The normal human keratinocyte cell line, HaCaT, was purchased from CLS Cell Lines Service GmbH (Eppelheim, Germany). Dulbecco’s modified Eagle’s medium (DMEM) was purchased from Capricorn Scientific GmbH (Ebsdorfergrund, Germany), fetal bovine serum (FBS) was purchased from GIBCO by Life Technologies (Paisley, UK), and trypsin-EDTA was purchased from Sigma-Aldrich (St. Louis, MI, USA). Cells were cultured in BioLite petri dishes (Thermo Fisher Scientific, Drive Rochester, NY, USA) in a humidified atmosphere with 5% CO_2_ at 37 °C in the incubator. Individual experiments were performed in 96-well plates (Thermo Fisher Scientific, USA). CellTiter 96^®^ AQ_ueous_ One Solution Cell Proliferation assay was purchased from Promega (USA), while measurement was performed on the microplate reader (Tecan, Switzerland). 2’,7’-Dichlorofluorescin diacetate (DCFH-DA), for the measurement of reactive oxygen species, was purchased from Sigma-Aldrich (St. Louis, MO, USA), while related analysis was performed on Varian Cary Eclipse Fluorescence Spectrophotometer equipped with a plate reader (Palo Alto, CA, USA).

Grape pomace was obtained from the Croatian native grape cultivar Kutjevo d.d., *Vitis vinifera* cv. Graševina, freeze-dried (Alpha 1-2 LD plus Christ, Osterode am Harz, Germany) for two days, milled and stored at 25 °C in a desiccator until the preparation of extracts.

Software BIOVIA COSMOtherm 2020 version 20.0.0. (Dassault Systemes, Paris, France) was used for the activity coefficient calculation of Graševina pomace polyphenols in DES.

Software BIOVIA TmoleX19 version 2021 (Dassault Systemes, Paris, France) was used for the geometric and energetic optimization of HBA, HBD and polyphenols used in this study.

### 3.2. COSMO-RS Simulations

To use COSMO-RS, the geometry and charge density of the individual molecules of a system need to be optimized using DFT. In this work, each molecule was optimized using the COSMO-BP-TZVP template of the TmoleX software package (interface of TURBOMOLE), which includes a def-TZVP basis set, DFT with the B-P83 functional level of theory, and the COSMO solvation model (infinite permittivity). All COSMO-RS calculations were performed using the software BIOVIA COSMOtherm 2020 version 20.0.0 with the BP_TZVP_C30_19.ctd parametrization. Quaternary ammonium salts applied as HBA were dealt as ion pairs and were then optimized using TmoleX. Organic acids were treated as protonated specimens. DES were treated as binary mixtures of HBD and HBA at a fixed stoichiometric rate within the framework of COSMO-RS. With this approach, a vast number of DESs are accessible without additional quantum chemical calculation, that is especially relevant for DESs screening. For any compound, its solubility in a solvent is inversely proportional to its activity coefficient in the system. As such, COSMOtherm was used to predict the activity coefficient of catechin in DESs at 60 °C and infinite dilution with 30% of water. 

### 3.3. DES Preparation 

DESs were prepared and characterized as described in Mitar et al. (2019) [36]. The DESs were prepared at certain molar ratios of betaine to hydrogen bond donor (HBD). The two components were placed in a specific ratio, with 30% (*v*/*v*) of water, in a 50 mL reagent bottle with a screw cap and were then placed in Shaker–Incubator ES-20/60 (Biosan, Riga, Latvia) stirred and heated to 50 °C for 2 h until a homogeneous transparent colorless liquid was formed. DES abbreviations and corresponding mole ratios are given in Table 3. 

### 3.4. Solid-Liquid Extractions

Extraction was performed as described in Cvjetko Bubalo et al. (2016) [39], in an ultrasound (US) bath XUB5 (XUB Series Digital Ultrasonic Baths, BioSan, Latvia) equipped with Digital LCD controls, a timer and a heater (heater power 150 W). All extractions were carried out under US (power 100 W) at a constant temperature (65 °C) for 50 min. Solid–liquid ratios of 0.5 g of freeze-dried grape pomace per 10 mL of prepared DES, which contained 30% (*v*/*v*) of water (BGlc, BScu and BEG) or aqueous ethanol (70% of ethanol), were used for extraction. Extracts were then centrifuged for 15 min at 5000 g, the supernatant was decanted, adjusted to a final volume of 10 mL (0.03 mg mL^−1^) and stored at +4 °C until further analyses were performed. 

### 3.5. Determination of Total Phenolic Content 

Total phenolic content (TP) was determined by the Folin–Ciocalteu method, as briefly described in Singleton et al. (1999). The absorbance was measured at 760 nm and results were expressed as mg of gallic acid equivalent per g of grape pomace (mg_GAE_ g^−1^). Spectrophotometric analyses were conducted in triplicate.

### 3.6. Oxygen Radical Absorbance Capacity Assay (ORAC) 

An oxygen radical absorbance capacity (ORAC) assay was performed based on the method described by Panić et al. (2019) and Mitar et al., (2019) [15,36], and the results were expressed as relative ORAC values (Equation (1)). Briefly, measurements were performed in 3 mL of reaction mixture with 2.25 mL of fluorescein sodium salt (0.04 μmol L^−1^) in sodium phosphate buffer (0.075 M, pH 7.0) and 0.375 mL diluted extracts, Trolox (25 μmol L^−1^) as standard or 0.075 M sodium phosphate buffer (pH 7) as a blank control. After incubation for 30 min at 37 °C 0.375 mL of AAPH was added. Fluorescence was recorded every minute up to value zero by a Varian Cary Eclipse Fluorescence Spectrophotometer (Palo Alto, CA, USA) with 485 nm excitation and 520 nm emission. Results were analyzed using the differences of areas under fluorescein decay curve between the blank and the sample. The results were the mean values (*n* = 3) and were expressed as µmol Trolox equivalent per g of extract (μmol TE g^−1^).

Relative ORAC value (μmol TE g^−1^) was calculated according to Equation (1): (1)$$\mathrm{relative}\;\mathrm{ORAC}-\mathrm{value}\;=\left(\frac{{\mathrm{AUC}}_{S^{-}}{\mathrm{AUC}}_{B}}{{\mathrm{AUC}}_{{\mathrm{TRX}}^{-}}{\mathrm{AUC}}_{B}}\right)\;\times \;k\;\times \;\alpha \;\times \;h,$$
where AUC_S_ is the area under the curve of sample, AUC_B_ is the area under the curve of blank, AUC_TRX_ is the area under the curve of Trolox, k is the dilution factor, α is the molar concentration of Trolox, and h is the ratio of the volume and mas of the sample. 

### 3.7. HPLC Analyses

Polyphenols were identified and quantified using an HPLC system (1260 Infinity II, Agilent, Santa Clara, CA, USA) coupled with a diode array detector (UV/DAD, 1260 Infinity II, Agilent, USA) and an automatic sampler (1260 Infinity II, Agilent, USA). Separation was achieved on a Poroshell 120 SB C18 column (150 mm, 4.6 mm, 5 µm; Agilent, USA) using H_2_O with 0.25% AcOH (A) and ACN (B) as the mobile phases. Gradient elution was modified as follows: 0–7.5 min 10% B, 7.5–15 min from 10% to 15% B and 15–25 min from 15% to 10% B. The flow rate was 1 mL min^−1^. The sample injection volume was 15 μL and the samples were always filtered through 0.22 μm polytetrafluoroethylene (PTFE) filters prior to injection. The column temperature was kept at 40 °C. UV-DAD acquisitions were carried out in the 200–600 nm range, while chromatograms were acquired at 280, 320 and 340 nm.

The retention times and spectral data of polyphenolic compounds were compared with external standards. Epigallocatechin, epicatechin, catechin, procyanidin B1, procyanidin B2, procyanidin B3, procyanidin B4, procyanidin C1 and flavan-3-ol derivatives were identified at 280 nm and rutin trihydrate at 360 nm. Polyphenols were quantified considering calibration curves of authentic external standards (10–1000 mg L^−1^) at the wavelength of maximum absorbance.

HPLC analyses were conducted in triplicate. Content of polyphenols was expressed as mg of compound per g of dry weight (dw).

### 3.8. Statistical Analysis 

Statistical analyses were performed using the software Statistica (Statsoft Inc., Tulsa, OK, USA), version 10. Where required, the measurements were processed using Tukey’s HSD test and statistical difference (*p* < 0.05) was designated by lower-case letters.

### 3.9. Determination of Antiproliferative Activity 

Antiproliferative activity of the grape pomace extracts prepared in DES and ethanol, as a referent solvent, were evaluated in vitro against adherent normal human keratinocytes HaCaT by the CellTiter 96^®^ AQ_ueous_ One Solution Cell Proliferation (MTS) assay. HaCaT cells were cultured in DMEM supplemented with 5% FBS and were maintained in BioLite petri dishes in the incubator with a humidified atmosphere and 5% CO_2_ at 37 °C. Individual experiments to test the cytotoxicity of the prepared extracts were performed in BioLite 96-wells plates seeded with exponentially growing cells at the concentration (~3 × 10^4^ cells per well in 100 µL of media) and incubated for 24 h, after which the treatment was administered. Grape pomace extracts (GPBGlc and GPEtOH) were diluted in the culture medium when applied to the cells, so the final volume ratio was 0.5%, 1.5%, 2.5% and 5% (*v*/*v*), while control cells were non-treated cells. Upon 72 h of treatment, the CellTiter 96^®^ AQ_ueous_ One Solution Cell Proliferation assay was performed according to the manufacturer’s instructions with minor modification. Briefly, 10 µL of the MTS reagent was added to each well, and cells were incubated for a further 3 h, after which absorbance at 490 nm was measured on the microplate reader. Cell viability was expressed as the percentage of treated versus control cells. The experiments were performed three times with five parallels for each volume ratio and data were expressed as the means ± S.D.

## 4. Conclusions

Here, the design of DES for obtaining ready-to-use extract from Graševina grape pomace was supported by computational screening and was compared with the experimental results, which confirmed COSMOtherm as an effective tool for developing a new tailor-made DES for the extraction of polyphenols by analyzing the activity coefficient. In a relatively short time, COSMOtherm software enabled us to quickly search for and select the optimal DES among 46 tested in silico, and to design an environmentally friendly extraction process for Graševina polyphenols. The prepared ready-to-use extract was characterized and it possesses desirable activity toward the growth of HaCaT cells.

Therefore, the newly developed product from wine waste could be recommended for use in the cosmetic industry. Moreover, by obtaining ready-to-use extract enriched with polyphenols, added value is created for the entire production of Graševina wine through an ecological and sustainable approach to waste management.

## Figures and Tables

**Figure 1 molecules-26-04722-f001:**
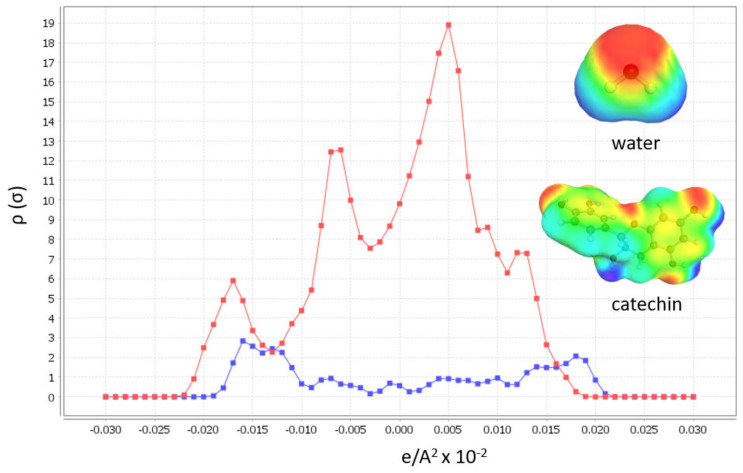
Sigma profiles of catechin (red) and water (blue) and their sigma surfaces.

**Figure 2 molecules-26-04722-f002:**
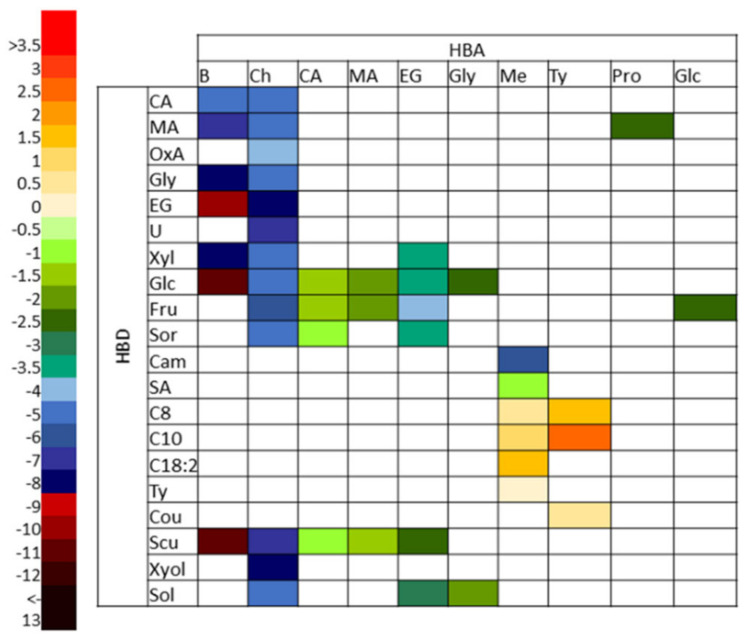
Predicted ln γ_solutes_ in DES at 60 °C using COSMO-RS. Labels: CA–citric acid; MA–malic acid; OxA–oxalic acid; Gly–glycerol; EG–ethylene glycol; U–urea; Xyl–xylose; Glc–glucose; Fru–Fructose; Sor—sorbose; Cam–camphour; SA–salicylic acid; C8–ctanoic acid; C10–decanoic acid; C18:2–linoleic acid; Ty–Thymol; Cou–coumarin; Scu–sucrose; Xyol–xylitol; Sol–sorbitol; B–betaine; Ch–Choline chloride; Me–menthol; Pro–prolin.

**Figure 3 molecules-26-04722-f003:**
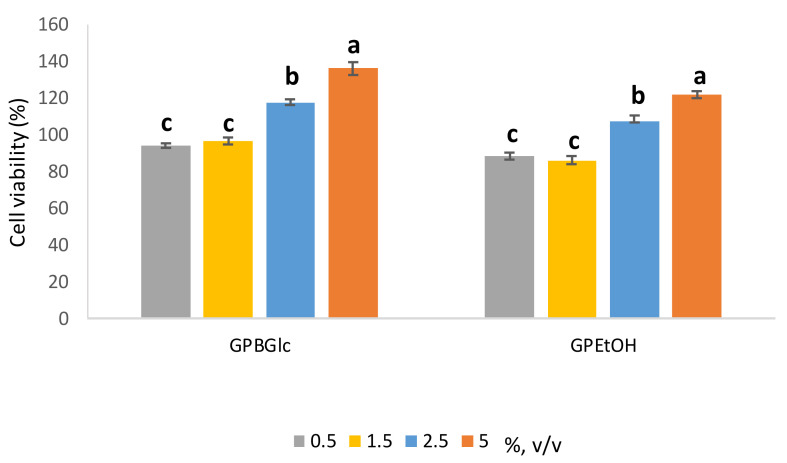
Effect of prepared extracts on HaCat cell viability determined by the MTS assay was assessed in volume ratio 0.5–5% (*v*/*v*). Cell viability (%) was expressed as percentage of treated cells versus control cells and the data from 3 individual experiments were expressed as the means (*n* = 4) ± S.D. Presented values followed by different lower-case letters (a–c) are significantly different (*p* < 0.05) as measured by Tukey’s HSD test.

**Table 1 molecules-26-04722-t001:** Molar ratios of HBD and HBA in DES tested in COSMOtherm software for catechin solubility.

DES	Molar Ratio	DES	Molar Ratio
B:CA	1:1	B:Glc	1:1
B:MA	1:1	Ch:Glc	1:1
Ch:CA	1:1	CA:Glc	1:1
Ch:MA	1:1	MA:Glc	1:1
Ch:OxA	1:1	EG:Glc	2:1
Pro:MA	1:1	Gly:Glc	2:1
B:Gly	1:2	Ch:Fru	1:1
Ch:Gly	1:2	CA:Fru	1:1
B:EG	1:2	MA:Fru	1:1
Ch:EG	1:2	EG:Fru	2:1
Ch:U	1:2	Glc:Fru	1:1
B:Xyl	1:1	Ch:Sor	1:1
Ch:Xyl	2:1	CA:Sor	2:3
EG:Xyl	2:1	EG:Sor	2:1
Me:C8	1:1	Me:Cam	1:1
Ty:C8	1:3	Me:SA	4:1
Me:C10	1:1	Me:C18:2	1:1
Ty:C10	1:1	Me:Ty	3:2
B:Scu	1:1	Ty:Cou	3:2
Ch:Scu	2:1	Ch:Xyol	5:2
CA:Scu	1:1	Ch:Sol	1:1
MA:Scu	1:1	EG:Sol	2:1
EG:Scu	2:1	Gly:Sol	2:1

Labels: CA–citric acid; MA–malic acid; OxA–oxalic acid; Gly–glycerol; EG–ethylene glycol; U–urea; Xyl–xylose; Glc–glucose; Fru–fructose; Sor–sorbose; Cam–camphour; SA–salicylic acid; C8–octanoic acid; C10–decanoic acid; C18:2–linoleic acid; Ty–Thymol; Cou–coumarin; Scu–sucrose; Xyol–xylitol; Sol–sorbitol; B–betaine; Ch–Choline chloride; Me–menthol; Pro–prolin.

**Table 2 molecules-26-04722-t002:** Polyphenolic content (mg g _dw_^−1^ of pomace) in prepared extract. Specific polyphenols contents and total polyphenols content were expressed as the means (*n* = 3) ± S.D.

Extracts ^1^					
Compound	GPBGlc	GPBScu	GPBEG	GPTyC10	GPEtOH
Epigallocatechin	-	-	-	-	0.2789
Flavan-3-ol derivative 1	0.4399	-	-	-	-
Procyanidin B1	0.3925	0.3270	-	0.5400	0.5049
Procyanidin B3	0.9145	0.1938	0.4182	-	1.1969
Flavan-3-ol derivative 2	0.3742	-	-	-	-
Catechin	0.6006	0.7589	0.5703	-	0.8198
Procyanidin B4	0.1793	0.2014	-	-	-
Procyanidin B2	0.2127	0.2175	0.1176	-	-
Epicatechin	0.5104	0.3928	0.4111	-	0.4714
Procyanidin C1	0.1139	0.0884	-	-	-
Flavan-3-ol derivative 3	0.2249	0.2470	-	-	-
Rutin	0.1214	0.0166	0.2318	-	0.2807
Total polyphenols ^2^	26.07 ± 0.16 ^a^	6.64 ± 1.04 ^c^	8.72 ± 1.52 ^b^	0.5 ± 0.07 ^d^	26.06 ± 1.59 ^a^

^1^ GPBGlc-grape pomace extract in DES BGlc (70%, *v*/*v*); GPBScu- grape pomace extract in DES BScu (70%, *v*/*v*); GPBEG-grape pomace extract in DES BSEG (70%, *v*/*v*); GPTy:C10-grape pomace extract in DES Ty:C10; GPEtOH-grape pomace extract in aqueous ethanol (70%, *v*/*v*). ^2^ presented value followed by different lower-case letters (a–d) are significantly different (*p* < 0.05) as measured by Tukey’s HSD test.

**Table 3 molecules-26-04722-t003:** Used DES in this research.

Deep Eutectic Solvents	Abbreviation	Molar Ratio
Betaine:Glucose	BGlc	1:1
Betaine:Sucrose	BScu	1:1
Betaine:Ethylene glycol	BEG	1:2
Thymol:decanoic acid	TyC10	1:1

## Data Availability

The authors confirm that the data supporting the findings of this study are available within the article.

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
