# Peer review of "COSMOtherm as an Effective Tool for Selection of Deep Eutectic Solvents Based Ready-To-Use Extracts from Graševina Grape Pomace"

_molecules, 2021, doi:10.3390/molecules26164722_

Round 1

Reviewer 1 Report

In this work, authors have carried out an exhaustive study regarding the application of different NADES for the extraction of polyphenols in Grasevina Grape pomace to obtain ready-to-use extracts for cosmetic applications. With this aim a theoretical initial study followed by extraction confirmation and biological characterization was developed. The work present great interest since other authors have already carried out the extraction of polyphenols from grape using DES the application of COSMOS selection considerable simplifies the procedure and reduce the cost. Based on that, from my point of view this work will reach the standards to be published and the journal Molecules after the following minor changes were amended.

Introduction:

  • Previous articles in which DES were used for the extraction of polyphenols from grapes should be included (Applied Sciences, 10, 2020, 4830; Bulletin of the Korean Chemical society, 41, 2020, 1175-1183).

Results and discussion:

  • Ethylene glycol is not a natural component to prepare NADES, please revise this issue.
  • How can explain the authors that the physicochemical properties of tested NADES do not have influence on the extraction process. Do the authors think that the kind of interaction analyte-NADES does not have influence?
  • Experimental characterization of the NADES should be carried out to assure the correct preparation. How can guarantee the authors that the solvents in correctly prepared?
  • For ORAC analyses, the experiments just with NADES should be carried out in ordet to accurately evaluate the influence of betaine on the final extract activity.
  •  

Experimental:

  • Lines 283-288: Not all NADES were prepared with betine, please revise this part of the text. Why did the authors decide to use 30% water? Was that based on previous studies? Are they published?
  • Regarding the extraction process, was it previously optimized? Was it based on previous studies? Please, specify that.

Reviewer 2 Report

The authors present an interesting work. The research aims to study the extraction and isolation of biologically active compounds from  Graševina  grape pomace using deep eutectic solvents.  I found that the manuscript to be lacking in the results and discussion part. Moreover, the paper is not very clearly written, and the English tends to be poor. Hence, I cannot recommend it to accept for publication.

  1. The author should elaborate on the introduction part.
  2. Why are the authors calling NDES and how they are different from normal DES?
  3. The resolution of Figures is very poor. Provide high-resolution images.
  4. The authors calculated the activity coefficient of polyphenols in different NDESs at a 1:1 molar ratio of HBA to HBD. Does all the investigated/screened DESs are liquid at a 1:1 molar ratio?
  5. Why don't the authors predict the DES curve of Thymol: C10 to check the reliability of COSMOtherm and compare it with the experimental?
  6. I do not agree with the authors that the COSMO-RS model is not suitable for the calculation of thermodynamic properties with ionic species of salt. There are lots of published articles that demonstrated the combined (C (cation) +A (anion)) and individual ‘C’ and ‘A’ approaches for predicting the thermodynamic properties using the COSMO-RS model.
  7. If the author used the combined C+A method for the COSMO-RS calculations, in such a case, the salt molecules exhibit lots of different conformers. How did the authors deal with different conformers while predicting the activity coefficients of polyphenols in DESs?
  8. The authors did not well execute the obtained results. There is a lot of underlying chemistry ignored especially with the COSMO-RS calculations.
  9. The manuscript is riddled with grammatical and syntactical errors. Literally, the whole manuscript contains multiple errors with respect to word usage, grammar, or syntax. The entire manuscript must be carefully proofread before resubmission.

Round 2

Reviewer 2 Report

The concerns I raised in the previous review are fixed now. I recommend acceptance of the manuscript.